# Selective Inhibition of PDE4B Reduces Methamphetamine Reinforcement in Two C57BL/6 Substrains

**DOI:** 10.3390/ijms23094872

**Published:** 2022-04-28

**Authors:** Kevin M. Honeywell, Eliyana Van Doren, Karen K. Szumlinski

**Affiliations:** 1Department of Psychological and Brain Sciences, University of California Santa Barbara, Santa Barbara, CA 93106-9660, USA; honeywell@ucsb.edu (K.M.H.); eliyanavandoren@ucsb.edu (E.V.D.); 2Department of Molecular, Cellular and Developmental Biology, University of California Santa Barbara, Santa Barbara, CA 93106-9660, USA

**Keywords:** PDE4, methamphetamine, sex differences, reinforcement, addiction, C57BL/6substrains

## Abstract

Methamphetamine (MA) is a highly addictive psychostimulant drug, and the number of MA-related overdose deaths has reached epidemic proportions. Repeated MA exposure induces a robust and persistent neuroinflammatory response, and the evidence supports the potential utility of targeting neuroimmune function using non-selective phosphodiesterase 4 (PDE4) inhibitors as a therapeutic strategy for attenuating addiction-related behavior. Off-target, emetic effects associated with non-selective PDE4 blockade led to the development of isozyme-selective inhibitors, of which the PDE4B-selective inhibitor A33 was demonstrated recently to reduce binge drinking in two genetically related C57BL/6 (B6) substrains (C57BL/6NJ (B6NJ) and C57BL/6J (B6J)) that differ in their innate neuroimmune response. Herein, we determined the efficacy of A33 for reducing MA self-administration and MA-seeking behavior in these two B6 substrains. Female and male mice of both substrains were first trained to nose poke for a 100 mg/L MA solution followed by a characterization of the dose–response function for oral MA reinforcement (20 mg/L–3.2 g/L), the demand-response function for 400 mg/L MA, and cue-elicited MA seeking following a period of forced abstinence. During this substrain comparison of MA self-administration, we also determined the dose–response function for A33 pretreatment (0–1 mg/kg) on the maintenance of MA self-administration and cue-elicited MA seeking. Relative to B6NJ mice, B6J mice earned fewer reinforcers, consumed less MA, and took longer to reach acquisition criterion with males of both substrains exhibiting some signs of lower MA reinforcement than their female counterparts during the acquisition phase of the study. A33 pretreatment reduced MA reinforcement at all doses tested. These findings provide the first evidence that pretreatment with a selective PDE4B inhibitor effectively reduces MA self-administration in both male and female mice of two genetically distinct substrains but does not impact cue-elicited MA seeking following abstinence. If relevant to humans, these results posit the potential clinical utility of A33 or other selective PDE4B inhibitors for curbing active drug-taking in MA use disorder.

## 1. Introduction

Globally, the use of the highly addictive psychomotor stimulant methamphetamine (MA) is a prevalent health issue. In the USA, MA use has reached epidemic proportions, with a 486.71% increase in the number of MA-positive urines detected during routine screening from 2013 to 2019 [1]. The reported lifetime prevalence of MA use is over 5% [2] with over 13% of all drug overdose deaths within the USA involving MA [3]. In addition to addiction, MA can induce a number of severe and debilitating neuropsychiatric conditions, including psychotic, affective, and cognitive disorders that can persist long following discontinuation of drug use, e.g., [4,5,6]. The neurobiological substrates underpinning the many psychiatric complications associated with MA use disorder are likely to involve multiple, potentially interacting mechanisms. However, one mechanism may involve alterations in the second messenger cyclic adenosine monophosphate (cAMP), resulting from changes in the activity of phosphodiesterase (PDE) 4B as repeated MA treatment increases PDE4B expression as a component of its neuroinflammatory response within the brains of laboratory rodents [7,8].

PDE4B is one of four members of the PDE4 family of PDE enzymes (PDE4A-D) that selectively deactivate cAMP [9,10]. PDE4 enzymes have received considerable experimental attention in animal models of substance use disorders [11,12,13,14,15,16,17,18], including MA use disorder [19,20,21,22,23] and in animal models of other neuropsychiatric conditions associated with prolonged MA use, including affective disorders [24,25,26,27,28], cognitive impairment/dementia [29,30], and schizophrenia [31] as well as various neurodegenerative diseases [32,33,34]. Supporting the translational potential of PDE4 inhibitors for treating substance use disorders, the non-selective PDE4 inhibitor ibudilast has advanced to Phase II clinical trials for MA use disorder (ClinicalTrials.gov (accessed on 13 February 2021) identifier: NCT01860807) and for oxycodone misuse (NCT01740414) and is currently in Phase 1 clinical trials for alcohol use disorder (NCT02025998) [35] with recent reports indicating efficacy and safety [36,37,38,39,40,41,42]. However, due to the lack of isozyme specificity, most centrally acting PDE4 inhibitors have accompanying tolerability issues associated with gastrointestinal dysfunction that have limited their advancement in many clinical trials [43,44]. Accordingly, there has been a concerted effort to generate isozyme-selective PDE4 inhibitors as tools for studying their therapeutic potential, and two studies to date indicate that selective inhibition of PDE4B is sufficient to reduce the self-administration of both cocaine [45] and alcohol [46] in laboratory rodents with no evidence for negative off-target effects.

Thus, the present study employed a murine model of oral MA self-administration to determine whether or not selective inhibition of PDE4B would be sufficient to reduce MA reinforcement and to attenuate a cue-elicited reinstatement of MA-seeking behavior. For this, we conducted a dose–response study of the selective PDE4B inhibitor A33 [47,48] on the maintenance of oral MA self-administration in mice and examined the effects of A33 pretreatment upon the capacity of a MA-paired cue to elicit drug-seeking behavior in MA-abstinent mice. A33 binds to a single amino acid in the C-terminus of PDE4B to promote the closing of this domain over the active site of the enzyme. Thus, A33 prevents cAMP binding to PDE4B and its hydrolysis [49,50]. A33 is 50-fold more selective for PDE4B (IC_50_ = 27 nM) than PDE4D (IC_50_ = 1569 nM) and other PDEs (IC_50_ > 10 μM) with a half-life of approximately 4 h in the mouse brain [49,50]. The behavioral data to date indicate that acute administration of A33 to laboratory rodents dose-dependently reduces depression-like signs without impacting anxiety-like measures or motor activity in mice [51], ameliorates contextual fear conditioning and deficits in spatial memory in a rat model of traumatic brain injury [52], and reduces binge drinking without affecting sucrose-drinking or sensitivity to the sedative-hypnotic properties of alcohol [46]. Thus, we hypothesized that A33 pretreatment would reduce oral MA reinforcement and attenuate cue-elicited MA seeking with little to no effect on sucrose reinforcement.

A sex difference exists in the cAMP/PDE4 response to the neuroimmune activator lipopolysaccharide (LPS) with females exhibiting lower PDE4B2, PDE4B3, and cAMP expression in the brain compared to males [53]. Whether or not the PDE4 isozyme profile of female mice contributes to the tendency of females to exhibit far greater behavioral sensitivity to MA, e.g., [54,55,56,57,58,59], is not known. To begin to address this question and further develop our murine model of oral MA reinforcement [60,61,62], the present study characterized the effects of A33 on the oral MA self-administration behavior of both male and female mice. As prior alcohol work indicated that the effects of A33 pretreatment generalize across the genetically distinct but related C57BL/6J (B6J) and C57BL/6NJ (B6NJ) substrains that differ in their alcohol intake [46], both substrains were included in our characterization of A33′s effects on oral MA reinforcement. For this, we conducted both dose- and demand-response studies in addition to an examination of cue-elicited MA seeking following a period of forced abstinence. Based on the extant literature and our prior results for alcohol [46], we hypothesized that MA would be more reinforcing in B6J and female mice but that A33 would effectively reduce MA reinforcement and seeking irrespective of sex or substrain.

As predicted, B6J and female mice did exhibit higher levels of MA reinforcement, but these group differences were either only apparent at moderately low MA concentrations (i.e., 50–400 mg/L) or during the early phases of MA self-administration procedures. Acute pretreatment with A33 did indeed reduce oral MA reinforcement in mice of both sexes, and substrains’ efficacy was detected at doses as low as 0.03 mg/kg. However, pretreatment with the highest A33 dose tested in this study (1.0 mg/kg) did not attenuate cue-elicited MA seeking. These data support the potential clinical utility of selective PDE4B inhibitors for interrupting active MA-taking but suggest that A33 may not be effective at attenuating cue-induced craving.

## 2. Results

### 2.1. Sex and Substrain Differences in Oral MA Reinforcement

To facilitate the navigation of the experimental design of our oral MA self-administration study, a schematic of the procedural timeline is provided in Figure 1 below.

#### 2.1.1. Acquisition

With the exception of two female B6J mice and one male B6NJ mouse, all mice reached our acquisition criteria for oral self-administration of a 100 mg/L MA reinforcer within the 30-day training period. Consequently, statistical analyses were conducted on data from 12 B6J males, 10 B6J females, 11 B6NJ males and 12 B6NJ females. Irrespective of sex, B6J mice required fewer days than B6NJ mice to reach the acquisition criteria (Figure 2A) (substrain effect: F (1,44) = 9.70, *p* = 0.003; sex effect and interaction, *p*’s > 0.35). A comparison of the average number of hole pokes into the “active” (MA-reinforced) hole over the last 3 days of self-administration training indicated more active nose poking in B6J versus B6NJ mice (substrain effect: F (1,44) = 7.98, *p* = 0.007) and in females versus males (Figure 2B) (sex effect: F (1,44) = 7.77, *p* = 0.008; interaction: *p* = 0.12). Females also emitted more inactive hole pokes than males over the last 3 days of self-administration training (Figure 2C) (sex effect: F (1,44) = 11.09, *p* = 0.002), but no substrain difference was detected for this measure (substrain effect and interaction, *p*’s > 0.41). To index the MA selectivity of responding as an additional index of MA reinforcement, we also compared the proportion of total responses that were directed at the active hole (response allocation). This comparison yielded a near-significant strain effect (F (1,44) = 4.01, *p* = 0.05) with B6J mice directing a greater proportion of their total response towards the active hole (Figure 2D), but no sex difference was detected for this variable (sex effect and interactions, *p*’s > 0.14). Consistent with the results for active hole poking (Figure 2A), females, on average, earned more 100 mg/L reinforcers over the last 3 days of self-administration than males (Figure 2E) (sex effect: F (1,44) = 7.43, *p* = 0.009) and consumed more MA (Figure 2F) (sex effect: F (1,44) = 8.28, *p* = 0.006). Irrespective of sex, B6J mice earned more 100 mg/L MA reinforcers than B6NJ mice (Figure 2E) (substrain effect: F (1,44) = 6.24, *p* = 0.02; Interaction: *p* = 0.12) and consumed more of this MA concentration (Figure 2F) (substrain effect: F (1,44) = 8.12, *p* = 0.007; Interaction: *p* = 0.17).

#### 2.1.2. Dose–Response Testing

The substrain differences in oral MA self-administration during the acquisition phase of this study prompted us to examine for sex and substrain differences in the dose–response function for oral MA reinforcement, and we did so across an approximately 100-fold dose range. As illustrated in Figure 3A, B6J mice appeared to respond in the active hole more than B6NJ when 50, 100, and 200 mg/L MA served as the reinforcer. A substrain difference in the shape of the dose–response function for active hole responding was supported by a significant substrain–concentration interaction (F (7,280) = 3.88, *p* < 0.0001). However, the magnitude of the substrain differences at these lower MA concentrations was such that *t*-tests corrected for 8 comparisons (α = 0.00625) did not detect significant differences at any of the specific MA concentrations (all *p*’s > 0.14). Although no sex differences or interactions were detected with respect to the MA dose–response function for active hole responding (sex effect and interactions, all *p*’s > 0.45), we detected a significant 3-way interaction for inactive hole responding (Figure 3B) (F (7,280) = 2.09, *p* = 0.047). Deconstructing the data along the substrain factor revealed greater inactive hole responding overall in B6J females versus males (Figure 3B, left) (sex effect: F (1,19) = 4.66, *p* = 0.048; concentration effect and interaction, *p*’s > 0.14). Although B6NJ females tended to respond more in the inactive hole than males at higher MA concentrations, no sex difference was apparent in this substrain (Figure 3B, right) (sex×concentration ANOVA, all *p*’s > 0.12). A substrain comparison of inactive hole responding within sex indicated a difference in the shape of the MA dose–response function in B6J versus B6NJ females rather than the absolute number of inactive responses at any given MA concentration (substrain × concentration: F (7,133) = 3.84, *p* = 0.001; post hoc *t*-tests, α = 0.00625, all *p*’s ≥ 0.05, n.s.). No substrain difference in inactive hole responding was detected in male subjects (substrain×concentration ANOVA, all *p*’s > 0.40). Perhaps not surprisingly, as operant-conditioning was well established, we detected no significant group differences in response allocation during MA dose–response testing (dose effect, *p* = 0.07; all other *p*’s > 0.10) with B6J mice exhibiting a non-significant trend towards higher response allocation (Figure 3C) (substrain effect: *p* = 0.07).

The pattern of group differences in the number of reinforcers earned during the MA dose–response phase of the study was similar to that observed for active hole responding (Figure 3D vs. Figure 3A) (substrain×concentration: F (7,280) = 2.76, *p* = 0.009; sex effect and interactions, all *p*’s > 0.06) with no statistically significant substrain differences detected at any of the MA reinforcer concentrations (*t*-tests, α = 0.00625, all *p*’s > 0.10). As the number of reinforcers earned was relatively stable across the different MA concentrations (Figure 3D), MA intake increased progressively and steeply with increasing reinforcer concentration (Figure 3E) (concentration effect: F (7,1280) = 57.41, *p* < 0.0001). Consistent with the very modest substrain difference in the number of MA reinforcers earned, we detected no statistically significant substrain or sex differences in the dose–response function for MA intake (Figure 3E) (substrain×sex×concentration ANOVA, other *p*’s > 0.16).

#### 2.1.3. Demand Testing

The relatively weak substrain differences in the dose–response function for MA reinforcement (Figure 3) prompted us next to examine whether or not sex or substrain differences might exist with respect to how much a mouse is willing to work for MA reinforcement when the response requirement is progressively increased across days. For this phase of testing, we held the MA reinforcer concentration at 400 mg/L as this concentration lay towards the start of the ascending limb of the dose-intake function (Figure 3E). Active hole responding declined significantly, albeit modestly, with increasing response requirement (schedule effect: F (3,120) = 2.80, *p* = 0.04). However, we detected no substrain or sex difference in this regard (Figure 4A) (substrain × sex × schedule ANOVA, other *p*’s > 0.15). In contrast to active hole responding, responding in the inactive hole did not vary with increasing response requirements (schedule effect and interactions, *p*’s > 0.11). Consistent with the dose–response phase of the study (Figure 3B), females emitted more inactive nose pokes than males during demand testing (Figure 4B) (sex effect: F (1,40) = 6.47, *p* = 0.015), but there was no evidence for any substrain difference in this measure (substrain effect and interactions, *p*’s > 0.20). Response allocation varied with increasing response requirement (schedule effect: F (13,120) = 2.82, *p* = 0.04), and there was a non-significant trend for a higher response allocation to the active hole by B6J versus B6NJ mice (Figure 4C) (substrain effect, *p* = 0.099); however, we detected no significant effects or interactions for this variable (substrain×sex×schedule ANOVA, other *p*’s > 0.54).

Given that the mice failed to adjust the number of nose pokes in response to the increasing response requirement, the number of MA reinforcers earned dropped precipitously with the increasing response requirement (schedule effect: F (3,120) = 101.1, *p* < 0.0001). As depicted in Figure 4D, there were no group differences in the number of reinforcers earned during demand-response testing (substrain×sex×schedule, all *p*’s > 0.18). Consistent with the drop in the number of reinforcers earned, MA intake also decreased with increasing response requirement (Figure 4E) (schedule effect: F (3,120) = 90.07, *p* < 0.0001) with no group differences detected (substrain×sex×schedule ANOVA, other *p*’s > 0.17).

### 2.2. A33 Effects on MA Reinforcement and Cue-Elicited Drug Seeking

#### 2.2.1. Maintenance of Established MA Self-Administration

When the mice had successfully acquired MA self-administration, we characterized the effects of A33 pretreatment (30 min earlier) on responding to and the intake of the 100 mg/L training dose of MA (see timeline in Figure 1). Irrespective of sex, B6J mice emitted more active nose pokes than B6NJ mice (substrain effect: F (1,41) = 7.82, *p* = 0.008; sex effect and interactions, *p*’s > 0.34), and A33 pretreatment dose-dependently lowered the number of active hole pokes in both sexes and substrains as indicated by a main A33 dose effect (F (4,164) = 17.41, *p* < 0.0001; A33 interactions, *p*’s > 0.63). Collapsing the data across sex and substrain, post hoc comparisons between active hole responding following VEH pretreatment and that following each of the A33 doses indicated a significant reduction at all four A33 doses (Figure 5A) (α = 0.0125 for four comparisons, 0 vs. 0.03 mg/kg: t(44) = 6.13, *p* < 0.0001; 0 vs. 0.1 mg/kg: t(44) = 7.04, *p* < 0.0001; 0 vs. 0.3 mg/kg: t(44) = 7.22, *p* < 0.0001; 0 vs. 1.0 mg/kg: t(44) = 4.96, *p* < 0.0001), and a test of within-subjects contrasts confirmed that the A33 effect on active hole responding was linear (F (1.41) = 25.05, *p* < 0.0001).

Despite the robust effect of A33 pretreatment on active hole responding (Figure 5A), none of the A33 doses altered inactive hole poking (Figure 5B) (A33 dose effect and interactions, all *p*’s > 0.20). Although an inspection of the data in Figure 5B suggested a more pronounced sex difference in inactive hole responding in B6J versus B6NJ mice, the interaction was not statistically significant (sex effect: F (1.41) = 18.60, *p* < 0.0001; substrain effect and interaction, *p*’s > 0.26).

In line with the results for active hole poking (Figure 5A), B6J mice earned more 100 mg/L MA reinforcers overall than B6J mice (substrain effect: F (1.41) = 9.36, *p* = 0.004), and all A33 doses reduced the number of reinforcers earned, relative to VEH (Figure 5C) (A33 dose effect: F (4.164) = 17.13, *p* < 0.0001; all other *p*’s > 0.19; post hoc comparisons: α = 0.0125, 0 vs. 0.03 mg/kg: t(44) = 6.77, *p* < 0.0001; 0 vs. 0.1 mg/kg: t(44) = 6.72, *p* < 0.0001; 0 vs. 0.3 mg/kg: t(44) = 7.41, *p* < 0.0001; 0 vs. 1.0 mg/kg: t(44) = 4.70, *p* < 0.0001). As observed for active hole responding, the A33-mediated reduction in the number of reinforcers earned was linear as indicated by a test of within-subjects contrasts (F (1.41) = 22.64, *p* < 0.0001). Although we did not detect a significant sex difference in the number of reinforcers earned (sex effect and interactions, *p*’s > 0.48), females consumed more of the 100 mg/L MA reinforcer overall than males (Figure 5D) (sex effect: F (1.41) = 4.48, *p* = 0.04). This sex difference appeared larger in B6J versus B6NJ mice (Figure 5D left vs. right); however, none of the interactions with the sex factor were statistically significant (sex interactions: *p*’s > 0.29). As expected based on the data for the number of reinforcers earned (Figure 5C), B6J mice also consumed more MA overall than B6NJ mice (Figure 5D left vs. right) (substrain effect: F (1.41) = 9.00, *p* = 0.005), and A33 pretreatment reduced MA intake at all doses tested regardless of sex or substrain (Figure 5D) (A33 dose effect: F (4.164) = 12.34, *p* < 0.0001; all other *p*’s > 0.73; post hoc tests, α = 0.0125; 0 vs. 0.03 mg/kg: t(44) = 5.00, *p* < 0.0001; 0 vs. 0.1 mg/kg: t(44) = 5.74, *p* < 0.0001; 0 vs. 0.3 mg/kg: t(44) = 7.04, *p* < 0.0001; 0 vs. 1.0 mg/kg: t(44) = 4.14, *p* < 0.0001). Additionally, the A33 effect on MA intake was also linear as indicated by a test of within-subjects contrasts (F (1.41) = 18.23, *p* < 0.0001), which is consistent with the data for the number of reinforcers earned.

#### 2.2.2. Cue-Elicited MA Seeking

A recent report indicated that repeated treatment with the non-selective PDE4 inhibitor roflumilast during forced abstinence from intravenous MA self-administration blunts the capacity of MA-paired cues to elicit conditioned responses in rats [29]. As acute A33 pretreatment is effective at reducing MA reinforcement (Figure 5) and binge drinking [53], we next determined whether or not acute pretreatment with A33, just prior to re-exposure to the MA-associated cues, would attenuate cue-elicited MA seeking following a period of abstinence. Upon the completion of the demand-testing phase of the study, mice were left undisturbed in their home cages for a period of 2 weeks. Then, mice underwent two sequential days of testing for cue-elicited responding after being pretreated with 0 or 1.0 mg/kg A33 30 min prior to the session (see Figure 1). The pretreatments were counterbalanced across the two days to control for order effects, and the inclusion of test order as a covariate did not influence the statistical outcomes of the data analysis. Overall, females emitted more active hole pokes than males during the final 2-day cue-testing phase of the study (Figure 6A) (sex effect: F (1,39) = 4.10, *p* = 0.05). Although an inspection of Figure 6A,B strongly suggested that this main sex effect reflected the higher responses of female versus male B6J mice, the sex–substrain interaction was not statistically significant (*p* = 0.11), and no other group differences were detected for this variable (substrain×sex×A33 dose ANOVA, all other *p*’s > 0.17). A comparison of response allocation in the active hole also failed to indicate any A33 pretreatment effect (A33 dose effect and interactions, p’s > 0.70); however, B6J mice allocated more of their responding in the active hole than B6NJ mice irrespective of sex (substrain effect: F (1,39) = 4.18, *p* = 0.048; sex effect and interactions, *p*’s > 0.41). Thus, A33 pretreatment does not significantly alter cue-reinforced responses following a period of forced abstinence. Females also emitted more inactive nose pokes than males during cue testing (Figure 6B) (sex effect: F (1,39) = 4.68, *p* < 0.0001), and we detected significantly higher inactive hole pokes by B6NJ versus B6J mice (substrain effect: F (1,39) = 6.79, *p* = 0.01). However, there was no indication of any A33 effect upon the responses in the inactive hole (A33 dose effect and interactions, all *p*’s > 0.3). These data suggest that pretreatment with 1.0 mg/kg A33 does not influence cue-reinforced responding following a period of MA withdrawal.

### 2.3. A33 Effects on Sucrose Reinforcement

To determine the selectivity of the A33 effect for MA self-administration behavior, we conducted a follow-up study in which female and male B6J mice were trained to nose poke for delivery of a 10% sucrose solution under a FR1 schedule of reinforcement over 7 days. Then, the effects of pretreatment with the highest and lowest A33 doses employed in the MA study (1.0 and 0.03 mg/kg, respectively) were determined on the maintenance of sucrose self-administration using a within-subjects design. This sucrose study employed B6J mice only as prior work indicated that 1.0 mg/kg A33 reduces sucrose drinking behavior in the home cage selectively in female B6J mice [54]. Thus, we sought to determine whether or not a sex-selective A33 effect might also be observed for B6J mice consuming sucrose under operant-conditioning procedures.

Not shown, both female and male B6J mice readily acquired the operant-conditioning procedures for sucrose reinforcement as indicated by a progressive increase and decrease, respectively, in the number of active versus inactive hole pokes over the course of the 7-day training period (for active hole pokes, day effect: F (6,108) = 15.084, *p* < 0.0001; for inactive hole pokes, day effect: F (6,108) = 26.06, *p* < 0.0001) as well as a progressive increase in the ratio of active hole responding (day effect: F (6,108) = 20.06, *p* < 0.0001), the number of reinforcers earned, and sucrose intake (for reinforcers, day effect: F (6,108) = 16.81, *p* < 0.0001; for intake, day effect: F (6,108) = 34.858, *p* < 0.0001). Females consumed more sucrose overall than males during the 7-day training period (sex effect: F (1,18) = 5.176, *p* = 0.035), but no other sex differences were detected during the acquisition phase of this study (sex effect for other variables, *p*’s > 0.23) as exemplified by the data averaged across the last 3 days of self-administration training (Figure 7).

In contrast to the linear reduction in MA self-administration behavior (Figure 5), the effect of A33 pretreatment on sucrose self-administration was bi-phasic in B6J mice with the 0.03 mg/kg dose increasing and the 1.0 mg/kg dose reducing active hole poking (Figure 8A) (A33 dose: F (2,40) = 19.76, *p* < 0.0001; sex effect and interaction, *p*’s > 0.35; post hoc *t*-tests (α = 0.05): 0 vs. 0.03 mg/kg, t(21) = 4.17, *p* < 0.0001; 0 vs. 1.0 mg/kg: t(21) = 4.50, *p* < 0.0001), the number of reinforcers earned (Figure 8D) (A33 dose: F (2,40) = 2171, *p* < 0.0001; sex effect and interaction, *p*’s > 0.07; post hoc *t*-tests (α = 0.05): 0 vs. 0.03 mg/kg, t(21) = 2.29, *p* = 0.03; 0 vs. 1.0 mg/kg: t(21) = 4.49, *p* < 0.0001), and sucrose intake (Figure 8E) (A33 dose: F (2,40) = 23.16, *p* < 0.0001; sex: F (1,20) = 11.39, *p* = 0.003; interaction, *p* = 0.22; post hoc *t*-tests (α = 0.05): 0 vs. 0.03 mg/kg, t(21) = 2.52, *p* = 0.02; 0 vs. 1.0 mg/kg: t(21) = 4.48, *p* < 0.0001). In contrast, the 0.03 mg/kg A33 dose did not alter responding in the inactive hole, while the 1.0 mg/kg dose lowered responses in both sexes (Figure 8B) (A33 dose: F (2,40) = 4.45, *p* = 0.02; sex effect and interaction, *p*’s > 0.89; post hoc *t*-tests (α = 0.05): 0 vs. 0.03 mg/kg, t(21) = 0.95, *p* = 0.35; 0 vs. 1.0 mg/kg: t(21) = 2.41, *p* < 0.03), and neither A33 dose influenced the allocation of responding in the active hole (Figure 8C) (sex×A33 dose ANOVA, all *p*’s > 0.69). These data indicate that while the capacity of high-dose A33 to lower MA reinforcement is non-selective, that of the minimally effective A33 dose (0.03 mg/kg) is not only selective but potentiated the reinforcing properties of the non-drug reinforcer sucrose.

## 3. Discussion

Here, we show that female mice from two distinct B6 substrains initially exhibit a higher level of oral MA self-administration when a moderately low (100 mg/L) MA solution serves as the reinforcer, but we failed to find evidence for a sex difference in sensitivity to MA reinforcement upon subsequent dose- and demand-response testing. We also show that a substrain difference also exists in oral MA self-administration with B6J mice exhibiting a higher level of MA reinforcement than B6NJ mice—an effect that appears also to manifest only at lower MA concentrations (<400 mg/L). However, irrespective of sex or substrain, pretreatment with the PDE4B isozyme-selective inhibitor A33 reduces established oral MA self-administration. The A33 effect on oral MA reinforcement is selective for MA-reinforced responding as (1) no A33 dose negatively impacted responding in the inactive hole and (2) the minimal effective A33 dose employed in our MA study (0.03 mg/kg) increased indices of sucrose reinforcement. These latter results, coupled with the failure of A33 pretreatment to reduce conditioned responding to MA-associated cues, argues that the capacity of A33 to reduce MA self-administration does not reflect off-target effects of the PDE4B inhibitor on sensorimotor, cognitive, or motivational processing.

The present data for MA extend our recent discovery that A33 effectively reduces binge alcohol drinking in both B6J and B6NJ mice without consistently impacting sucrose-drinking or spontaneous or alcohol-induced changes in motor co-ordination or activity [46], which are consistent with earlier indications that A33 doses less than 3 mg/kg produced no signs of motor impairment in either rats or mice [50,51]. The present MA data also align with the results of a recent cocaine study in B6J mice in which a novel selective PDE4B inhibitor, KVA-D-88, was found to reduce cocaine-induced locomotor hyperactivity and sensitization and cocaine reinforcement under both fixed and progressive ratio schedules of reinforcement [45]. Finally, our data extend over a decade’s worth of evidence indicating that non-selective PDE4 inhibitors attenuate the discriminative stimulus [22], locomotor-activating and locomotor-sensitizing [20] as well as reinforcing [19,21] properties of MA in rodents, which serves, in part, as a base for their investigation as pharmacotherapies for treating MA use disorder in humans [Refs. [37,38,39,41]; ClinicalTrials.gov (accessed on 13 February 2021) identifier: NCT01860807].

### 3.1. Selective PDE4B Inhibition by A33 Reduces MA Reinforcement

In this study, A33 pretreatment reduced MA reinforcement and intake with significant effects detected at the lowest dose tested (0.03 mg/kg; Figure 5). As reported for binge drinking [46], the dose–response functions for active nose poking, MA reinforcers earned and MA intake were relatively flat, even though the results of within-subjects contrasts indicated that the A33 dose–response functions for all of these variables were linear (Figure 5). We did not assay the effects of higher A33 doses in the present study based on evidence that doses greater than 3 mg/kg can induce off-target effects, at least in mice of the ICR strain [47]. Thus, it remains to be determined whether A33 doses higher than 1.0 mg/kg would exert a more robust effect on MA reinforcement and intake than observed herein. However, given that pretreatment with 1.0 mg/kg A33 significantly reduced indices of sucrose reinforcement in the present study (Figure 7), it is likely that doses higher than those tested herein would result in more pronounced non-selective effects on operant responding. Nevertheless, A33 doses ranging from 0.1–1.0 mg/kg were sufficient to reduce oral MA reinforcement by at least a third in mice of both sexes and both B6 substrains (Figure 5) with the lowest effective A33 dose (0.03 mg/kg) failing to alter sucrose reinforcement in B6J mice (Figure 7). Repeated MA exposure induces a pro-inflammatory response [63,64,65,66,67,68,69], including an increase in PDE4B expression [7,8]. Although we did not measure the expression of PDE4B or other pro-inflammatory markers herein, our data provide novel evidence supporting the hypothesis that MA-induced increases in PDE4B drive MA reinforcement. One psychopharmacological mechanism through which this might occur may involve a reduction in the subjective or interoceptive effects of the drug. In support of this notion, the non-selective PDE4 inhibitor rolipram reduces the discriminative stimulus properties of MA in rats [23], while another non-selective inhibitor ibudilast blunts MA’s subjective effects in humans [42]. Now that we have characterized oral MA reinforcement across a very wide range of MA concentrations and found that, under our operant-conditioning procedures, mice are capable of consuming very high amounts of a drug (~30 mg/kg/day) (Figure 3), future work can determine how A33 pretreatment might shift the dose–response functions for MA reinforcement in both the presence and absence of MA-associated cues in relation to its effects on the neuroimmune response to voluntary high-dose MA self-administration.

### 3.2. Selective PDE4B Inhibition by A33 Does Not Alter Cue-Elicited MA Seeking

In contrast to its effect on MA self-administration behavior, acute pretreatment with the highest A33 dose tested in this study (1.0 mg/kg) did not alter responses to MA-associated cues following a period of forced abstinence (Figure 6). This result is contrary to published studies employing repeated treatment with non-selective PDE4B inhibitors in which inhibitor pretreatment blunted stress- and drug-primed [21] or cue-elicited [22] MA-seeking behavior. While our negative results suggest that selective PDE4B inhibition is insufficient to alter cue-elicited MA-craving, a number of other potential explanations might account for the failure to detect an A33 effect on the reinforcing properties of MA-associated cues. First, in an effort to more fully characterize the oral MA self-administration phenotype of B6 substrains, the mice in this study underwent a relatively complex battery of operant conditioning, which included modifications of the dose of the MA reinforcer and the schedule of reinforcement (Figure 1) that likely impacted the conditioned reinforcing properties of the MA-associated cue, perhaps masking any potential A33 effects. Thus, future work will employ a more traditional experimental design in which mice undergo daily self-administration sessions, using a fixed MA concentration and reinforcement schedule prior to abstinence and testing for cue-elicited responding. Second, mice in this study had a history of repeated A33 treatment prior to the test for cue-elicited responding. Thus, it is possible that tolerance developed to the reinforcement-attenuating effects of A33. While we did not assay the effects of repeated treatment with a fixed dose of A33 on MA self-administration herein, we demonstrated previously that the capacity of 1.0 mg/kg A33 to reduce binge drinking is unchanged in either B6J or B6NJ mice following 5 injections, spaced 3 days apart [46]. While such findings argue against the development of tolerance as a likely contributor to the negative results for cue-elicited MA seeking, this possibility cannot be ruled out at the present time.

Related to this, a recent study demonstrated that repeated treatment with the non-selective PDE4 inhibitor roflumilast early during a period of forced abstinence is sufficient to attenuate both subsequent cue-elicited MA seeking and the resumption of MA self-administration in rats [22]. Thus, it is possible that repeatedly inhibiting PDE4, or possibly PDE4B specifically, during early MA withdrawal may be a more effective strategy for curbing subsequent drug-craving than administering such inhibitors prophylactically later in drug abstinence (i.e., >2 weeks post-MA). However, arguing against this notion are data indicating that three daily injections of the non-selective PDE4 inhibitor ibudilast, starting 10 days following the last MA self-administration, are effective at reducing both stress- and MA-primed reinstatement of MA seeking in rats that had undergone extinction procedures prior to testing [21]. How the precise animal model of MA-craving/relapse employed (e.g., extinction-reinstatement versus forced abstinence) impacts the effects of PDE4/4B inhibition on study outcomes is currently not known. As the issues of tolerance and timing of repeated inhibitor dosing have obvious clinical implications, it will be important in future work to study these factors and relate “therapeutic efficacy” to the biochemical effects of MA (e.g., changes in neuroinflammatory or monoaminergic/cAMP signaling) more systematically.

### 3.3. Substrain Differences in Oral MA Reinforcement

Although B6J mice are reported to exhibit lower self-administration of low-concentration MA solutions (e.g., 20–40 mg/L) than the completely unrelated DBA/2J substrain, e.g., [70,71]—a finding linked to a mutation in the gene encoding the trace amine-associated receptor 1(TAAR1) that is non-functional in DBA/2J mice [72,73]—we have shown repeatedly that male mice on a B6J background will readily consume MA concentrations greater than 40 mg/L under operant-conditioning procedures [61,62]. Herein, we expanded our characterization of the MA self-administration phenotype of B6J mice to include a dose–response study across a 100-fold dose range (40 mg/L to 3.2 g/L), and we compared the phenotype of B6J mice to that of the related, but genetically distinct, B6NJ substrain. The B6NJ and B6J substrains diverged nearly 70 years ago and exhibited different behavioral responses to several different drugs of abuse, e.g., [74,75,76,77,78], divergent binge eating [79] and drinking [46] phenotypes. Herein, the 100 mg/L MA concentration was clearly more reinforcing in B6J versus B6NJ mice—a substrain difference apparent during both the acquisition (Figure 2) and maintenance (A33 testing; Figure 5) phase of this study. Relative to B6NJ mice, B6J mice also tended to exhibit higher levels of MA reinforcement but not MA intake at concentrations between 50 and 200 mg/L MA (Figure 3); however, no substrain differences were detected in either MA reinforcement or intake at MA concentrations higher than 200 mg/L (Figure 3) regardless of the reinforcement schedule (Figure 4). It is tempting to speculate at this time that B6J mice may be more sensitive than B6NJ to the positive interoceptive/subjective effects of MA and that this increased sensitivity might motivate B6J to respond more robustly at relatively low MA concentrations or during earlier phases of self-administration procedures. However, the fact that the magnitude of the A33 effect on indices of MA reinforcement was identical between B6J and B6NJ mice (Figure 5) argues against substrain differences in sensitivity to or the efficacy of A33 for reducing MA reinforcement.

As we have reported previously for low-dose (e.g., 40 mg/L) oral MA [61,62] and oral fentanyl [80], neither the B6J nor the B6NJ mice in the present study adjusted their nose-poking behavior to compensate for an increase in the response requirement for reinforcement by 400 mg/L MA, and, consequently, MA intake dropped precipitously across increasing reinforcement schedules (Figure 4). Our collection of oral drug self-administration data to date argues that employing this particular within-subjects approach is ineffective at indexing drug demand or motivation for a drug under oral self-administration procedures. Thus, future work will incorporate more classic within-session progressive ratio schedules of reinforcement to index substrain and sex differences in the motivation to orally self-administer MA as well as the effects of A33 pretreatment therein.

### 3.4. Sex Differences in Oral MA Reinforcement

We detected no sex difference in the number of days required by mice to reach the criteria for the acquisition of MA self-administration (Figure 2A). However, female mice of both substrains exhibited higher oral MA reinforcement and intake than males during the acquisition period of testing when 100 mg/L MA served as the reinforcer (Figure 2). Curiously, we detected no other sex differences in MA reinforcement or intake once the mice entered the A33 testing phase of the study. The lack of sex differences in oral MA reinforcement, particularly during MA dose–response testing and on the tests for cue-elicited responding, was surprising as female rats intravenously self-administered larger amounts of MA than males and exhibited more robust reinstatement of MA seeking [54,55,56,57,58]. Consistent with published rat studies, e.g., [58], the female mice in our study exhibited a higher level of nose poking in the inactive hole than males, and this sex difference was observed consistently across all MA self-administration phases of our study. However, sex differences were not observed with respect to the allocation of responses between the active and inactive holes at any time during testing, indicating no sex difference in the MA selectivity of responding but rather a general increase in MA-induced motor hyperactivity. Indeed, female rodents are well-characterized to be more sensitive to the locomotor-activating properties of MA, e.g., [81,82], and we did not detect significant sex differences in inactive hole-poking behavior in the tests for cue-elicited MA seeking when MA was no longer available (Figure 6).

There are many explanations as to why females are more sensitive than males to the reinforcing and psychomotor-activating properties of MA, which include (but are certainly not limited to) genetic and activation effects of gonadal hormones, c.f. [83,84,85], and pharmacokinetics [81]. An additional factor may also be innate neuroimmune function, including PDE4B expression [86]. Females exhibit higher PDE4B expression and lower cAMP expression than males in cardiac myocytes [87]; however, no sex differences in basal mRNA expression of different PDE4B splice variants occur in the brain [53] (at least within the brain regions studied: hippocampal subregions, caudate putamen, cingulate cortex). However, particularly considering that MA is a potent activator of the immune system [63,64,65,66,67,68,69], females are reported to exhibit more rapid and robust changes in brain PDE4B2 and PDE4B3 messenger ribonucleic acid (mRNA) expression following an LPS-induced challenge of the innate immune system [53]. Determining whether or not a similar sex difference exists for a MA-induced challenge of the innate immune system is highly warranted given the critical role played by immune system activation in MA-induced neurotoxicity and the risk of neurocognitive and neurodegenerative disorders that have strong sex biases (e.g., Alzheimer’s disease and depression). The fact that our sex differences in MA reinforcement disappeared upon and following repeated A33 treatment begs the question as to whether or not repeated A33 administration might have exerted a greater effect on the MA reactivity of the neuroimmune system in females than in males, thereby eliminating sex differences in MA self-administration behavior?

## 4. Materials and Methods

### 4.1. Subjects

Adult (8–10 weeks of age) female and male B6J mice (catalog no. 000664; *n* = 24 with 12 females and 12 males) and B6NJ mice (catalog no. 005304; *n* = 24 with 12 females and 12 males) were obtained from The Jackson Laboratory (Sacramento, CA, USA). Mice were housed in same-sex groups of 4 and allowed a minimum of 7 days to acclimate to a climate- and humidity-controlled colony room under a reverse 12 h light/dark cycle (lights off at 11:00 h). Mice were identified using tail markings. Food and water were available ad libitum except during the 1 h MA self-administration period (see below). All the cages were lined with sawdust bedding, nesting materials, and an igloo in accordance with vivarium protocols. All experimental procedures were in compliance with The Guide for the Care and Use of Laboratory Animals (2014) and approved by the Institutional Animal Care and Use Committee of the University of California, Santa Barbara (protocol number 829.3).

### 4.2. Drugs

A33 (CAS number 121604-72-6) was purchased from Tocris Bioscience (Minneapolis, MN, USA) and was suspended in 10% dimethyl sulfoxide (DMSO) at a concentration of 100 mg/mL and sonicated for 45 min. The suspension was then diluted using saline to the final concentrations of 0.03, 0.1, 0.3, and 1.0 mg/mL (0.1% DMSO) with the 1.0 mg/mL concentration requiring additional sonication for complete dissolution. The vehicle (VEH) solution consisted of 0.1% DMSO in saline. A33 pretreatment occurred 30 min prior to behavioral testing and was administered intraperitoneally (IP) at an injection volume of 10 mL/kg. MA was purchased from Sigma-Aldrich (now MilliporeSigma; St. Louis, MO, USA) and was dissolved in potable water to final concentrations of 20 mg/L to 3.2 g/L for oral self-administration.

### 4.3. Operant Conditioning for Oral MA Reinforcement

Operant conditioning for oral MA reinforcement was conducted in 4 cohorts of 12 mice, starting at approximately 10:30 h each day. On each self-administration day, mice were relocated in their home cages to a non-colony procedural room approximately 30 min before testing. Mice were tested in 59.69 cm × 40.64 cm operant chambers (Med Associates, St. Albans, VT, USA) housed in a sound-attenuated cabinet and equipped with two nose poke holes (one inactive and one active/MA-reinforced) and a liquid receptacle attached to an infusion pump (Med Associates, St. Albans, VT, USA), which delivered 20 µL of the MA solution into the receptacle. Mice were allotted 30 days to acquire oral self-administration of 100 mg/L of the MA solution under a fixed ratio (FR) 1 schedule of reinforcement. The 100 mg/L reinforcer concentration was selected for study based on the results of published work indicating that this concentration lies on the ascending limb of the dose–response function for MA reinforcement [59,60] in B6J mice. Each reinforcer delivery was signaled by a compound light/tone cue and was followed by a 20 s time-out period. Mice were trained daily during a 1 h session until they reached the acquisition criterion of a minimum of 10 nose pokes/session with 3 consecutive days of stable responding and at least 70% of the nose pokes occurring in the active hole each session. Two female B6J mice (*n* = 2) and one male B6NJ mouse (*n* = 1) failed to acquire oral self-administration under this paradigm and thus were excluded from the statistical analyses of the data.

### 4.4. A33 Dose Response

As illustrated in Figure 1, following the 30-day acquisition phase, mice were randomly injected intraperitoneally with one of the following doses of A33: 0.03 mg/kg, 0.10 mg/kg, 0.30 mg/kg, 1.00 mg/kg or VEH (0.1% DMSO in saline), 30 min prior to the self-administration session. These A33 doses and pretreatment intervals are consistent with those employed in our recent study on binge drinking in which pretreatment with all A33 doses significantly reduced alcohol intake in both B6J and B6NJ mice [46]. The order of A33 dosing was administered in a pseudo-randomized fashion, ensuring that an approximately equal number of mice from each sex and substrain received each A33 dose on a given test day. To examine for carry-over effects, after A33 testing, each mouse was allowed one or more days to return to baseline responding (i.e., pre-testing levels; see Figure 2) and then were administered the next pseudo-randomly selected dose until all mice received all five A33 doses.

### 4.5. MA Dose- and Demand-Response Testing

A minimum of two days following completion of the A33 dose–response phase of the study, we determined the substrain and sex differences in the dose–response function for MA reinforcement. Similar to prior studies, e.g., [60,61,62], mice were presented with different concentrations of MA—the concentrations ranged from 25 mg/L to 3.2 g/L in this study—with each concentration available over five consecutive sessions. The average data for the last three sessions under each concentration was employed in the statistical analyses. Mice responded under an FR1 schedule throughout the MA dose–response testing. As illustrated in Figure 1, we next examined sex and substrain differences in the willingness to work for MA reinforcement by progressively increasing the number of nose pokes required by mice to earn each 20 µL reinforcer across days. For this phase of testing, we increased the MA reinforcer concentration to 400 mg/L—a concentration that lay at the base of the ascending limb of the dose-intake function in our animals in this study (Figure 3D). Mice were required to respond under each of the reinforcement schedules (FR1, FR2, FR5 and FR10) for three consecutive days. As intake dropped precipitously with increasing demand, we did not assay for group differences under higher reinforcement schedules.

### 4.6. Cue-Elicited MA Seeking

Following the last day of demand-response testing, mice were returned to the colony room, where they remained for 2 weeks. Following the 2-week period of forced abstinence, mice underwent two consecutive days of testing for cue-elicited responding, 30 min prior to which mice were injected with either 0 or 1.0 mg/kg A33. The order of A33 dosing was counterbalanced across animals within each substrain and sex to avoid order effects. During these tests for cue-elicited MA seeking, mice were placed into the operant chamber for 1 h. As conducted in prior studies, e.g., [60], nose poking in the active, previously MA-reinforced hole resulted in the presentation of the light/tone compound stimulus only (i.e., no MA was available), while responding in the inactive hole continued to have no consequences.

### 4.7. Data Analyses

The data were analyzed using mixed-factor ANOVAs with the between-subjects factors of substrain and sex and, depending on the phase of the study, the within-subjects factors of A33 dose (5 levels for dose–response; 2 levels for cue-elicited MA seeking), MA concentration (8 levels), or FR schedule (4 levels). Significant interactions were deconstructed along the factor(s) of interest, followed by post hoc *t*-tests using the Bonferroni correction for multiple comparisons, when appropriate.

## 5. Conclusions

A moderately low MA concentration (100 mg/L) is more reinforcing in female than male mice during early drug experience with female mice tending to exhibit more non-specific responses than males, particularly at higher MA concentrations or in response to MA-associated cues. Irrespective of sex, the PDE4B-selective inhibitor A33 effectively reduced oral MA reinforcement by mice of distinct B6 substrains that vary in their oral self-administration of moderately low MA concentrations. The capacity of A33 to blunt oral MA reinforcement did not relate to off-target motor effects in any obvious manner, and A33 did not affect responses in the inactive lever nor did it blunt the capacity of MA-associated cues to elicit conditioned responding. Although the 1.0 mg/kg A33 dose exerted a non-selective reduction also in sucrose reinforcement, the minimal effective A33 dose (0.03 mg/kg) selectively reduced MA reinforcement. These data provide initial evidence to support the potential clinical utility of selectively targeting PDE4B for treating active drug taking in MA use disorders but do not support its utility in modulating MA craving during drug abstinence.

## Figures and Tables

**Figure 1 ijms-23-04872-f001:**
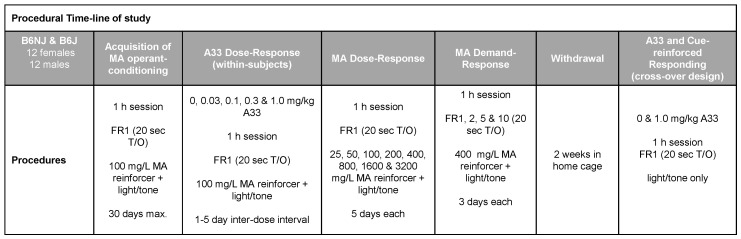
Summary of the procedural timeline for the MA study, which employed a within-subjects design to determine (1) sex and substrain differences in oral methamphetamine (MA) reinforcement and (2) the dose–response function for A33 effects upon oral MA reinforcement and the effect of 1.0 mg/kg A33 on cue-reinforced MA-seeking behavior.

**Figure 2 ijms-23-04872-f002:**
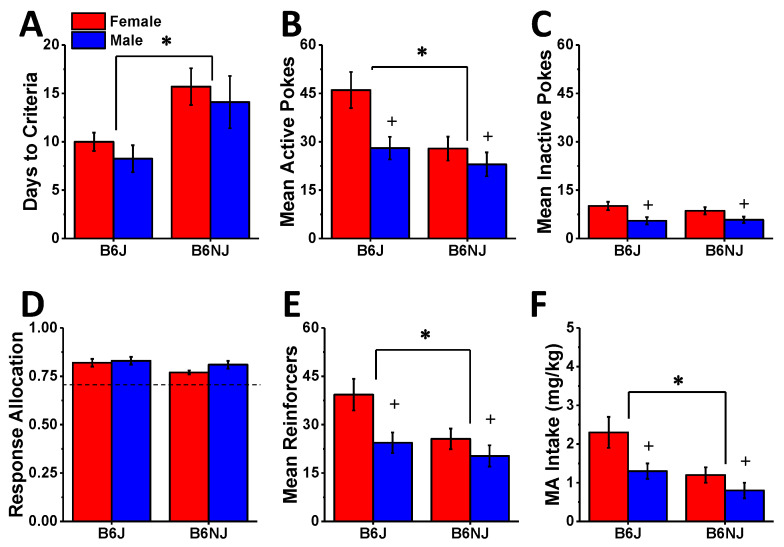
Summary of the substrain and sex differences in oral MA reinforcement during the acquisition phase of this study, highlighting the group differences in the number of days to reach the acquisition criterion (**A**) as well as the mean number of nose pokes into the active, MA-reinforced, hole (**B**), the inactive, non-reinforced hole (**C**), the relative allocation of responding in the active vs. inactive hole (**D**), the mean number of reinforcers earned (**E**), and the MA intake (**F**) over the last 3 days of self-administration. The data represent the mean ± SEM of 10–12 mice/sex/substrain. * represents a significant substrain effect, *p* < 0.05; + represents a significant sex effect, *p* < 0.05.

**Figure 3 ijms-23-04872-f003:**
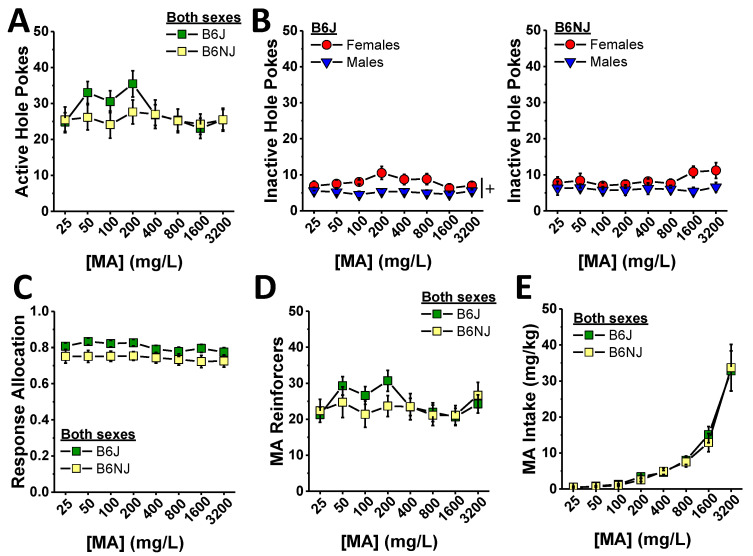
Summary of the substrain and sex differences in MA reinforcement during the dose–response phase of the study, highlighting the small or no group difference in the number of nose pokes into the active, MA-reinforced, hole (**A**) and the inactive, non-reinforced hole (**B**) in B6J (left) and B6NJ mice (right) as well as the response allocation (**C**), the number of reinforcers earned (**D**), and the MA intake (**E**) over a ~100-fold range in MA reinforcer concentrations. The data represent the mean ± SEM of 12 mice/sex/substrain. + represents a significant sex effect, *p* < 0.05.

**Figure 4 ijms-23-04872-f004:**
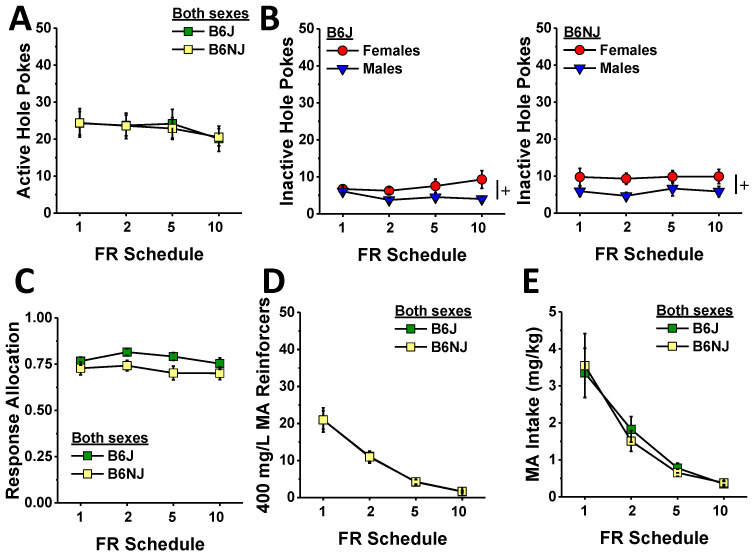
Summary of the substrain and sex differences in MA reinforcement during the demand-response phase of the study, highlighting the small or no group difference in the number of nose pokes into the active, MA-reinforced hole (**A**) and the inactive, non-reinforced hole (**B**) in B6J (left) and B6NJ mice (right) as well as the ratio of responses made in the active vs. inactive hole (**C**), number of reinforcers earned (**D**), and the MA intake (**E**) across the increasing response ratios. The data represent the mean ± SEM of 12 mice/sex/substrain. + represents a significant sex effect, *p* < 0.05.

**Figure 5 ijms-23-04872-f005:**
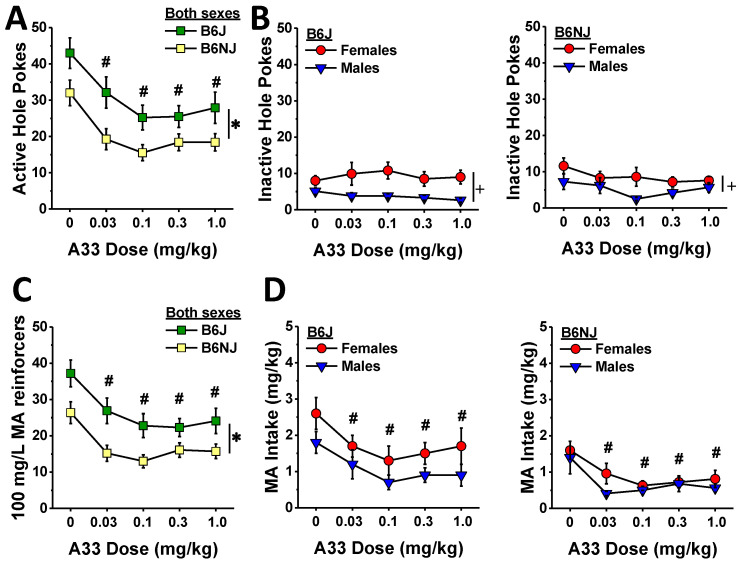
Summary of the effects of A33 pretreatment on established MA self-administration for a 100 mg/L MA reinforcer, highlighting the A33-induced reduction in the number of nose pokes in the active, MA-reinforced hole (**A**) but no A33 effect on nose poking in the inactive, non-reinforced hole (**B**) in B6J (left) and B6NJ mice (right). A33 also reduced the number of reinforcers earned, irrespective of sex (**C**) and lowered MA intake (**D**) in both B6J (left) and B6NJ (right) mice. The data represent the mean ± SEM of 12 mice/sex/substrain. * represents a main substrain effect; + represents a significant sex effect, *p* < 0.05; # *p* < 0.05 vs. 0 mg/kg A33 for both substrains/sexes.

**Figure 6 ijms-23-04872-f006:**
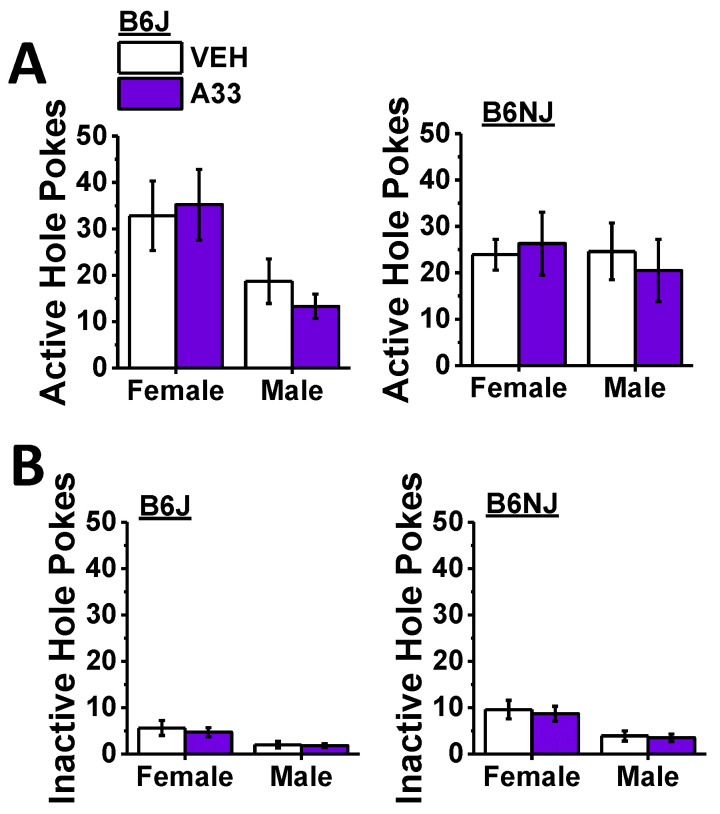
Summary of the effects of A33 pretreatment on cue-elicited responding following a 2-week period of forced abstinence, highlighting no effect of the inhibitor on the number of nose pokes into the active, MA-reinforced hole (**A**) or the inactive, non-reinforced hole (**B**) in B6J (left) and B6NJ mice (right). The data represent the mean ± SEM of 12 mice/sex/substrain.

**Figure 7 ijms-23-04872-f007:**
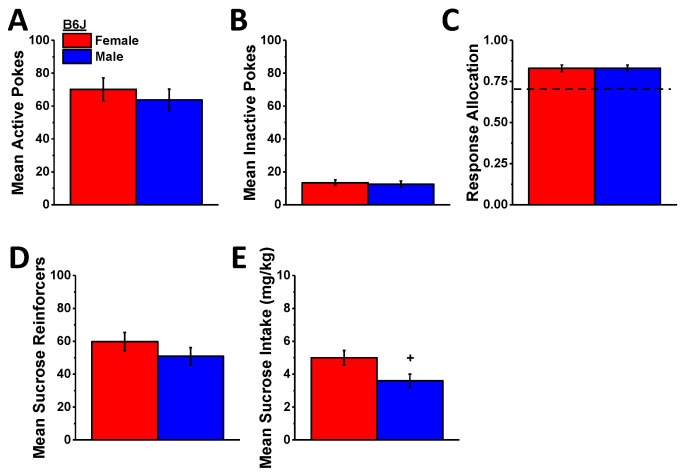
Comparison of oral 10% sucrose reinforcement between female and male B6J mice during the acquisition phase of this follow-up study, highlighting no sex differences in the mean number of nose pokes into the active, sucrose-reinforced hole (**A**) and the inactive, non-reinforced hole (**B**), the relative allocation of responding in the active vs. inactive hole (**C**), or the mean number of reinforcers earned (**D**). However, female B6J mice did consume more sucrose than males (**E**) over the last 3 days of sucrose self-administration training. The data represent the mean ± SEM of 11 mice/sex. + represents a significant sex effect, *p* < 0.05.

**Figure 8 ijms-23-04872-f008:**
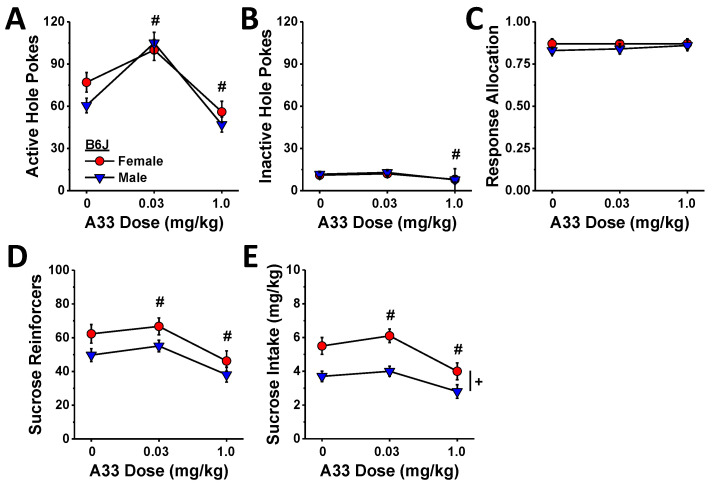
Summary of the effects of A33 pretreatment on established sucrose self-administration for a 10% sucrose reinforcer in B6J mice, highlighting a biphasic effect of A33 pretreatment on the number of nose pokes into the active, sucrose-reinforced hole (**A**), the number of reinforcers earned (**D**), and sucrose intake (**E**) by both female and male mice. Only the 1.0 mg/kg A33 dose reduced the number of nose pokes into the inactive, non-reinforced hole (**B**), and neither dose affected the allocation of responses across the two holes (**C**). The data represent the mean ± SEM of 11 mice/sex. + represents a significant sex effect, *p* < 0.05; # *p* < 0.05 vs. 0 mg/kg A33 for both sexes.

## Data Availability

The data presented in this study are available on request from the corresponding author.

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
