# Peer review of "Selective Inhibition of PDE4B Reduces Methamphetamine Reinforcement in Two C57BL/6 Substrains"

_ijms, 2022, doi:10.3390/ijms23094872_

Round 1

Reviewer 1 Report

The current study reports regarding methamphetamine self-administration in male and female mice of the two mouse substrains (B6J and BN6J). The authors found that female and male mice of B6J learn faster to self-administer than mice of the B6NJ substrain. Female mice of B6J also performed nose poke to a greater level than male mice of B6J and both male and female mice of B6NJ. In addition, the authors showed that administration of A33, a PDE4B inhibitor, reduced methamphetamine self-administration at all doses tested but there was no dose-response, i,e., at higher doses A33 did not further reduce methamphetamine self-administration. A33 also did not alter cue-mediated methamphetamine self-administration. Overall, the results are interesting and the manuscript is well written except the issues noted below.

Major:

  1. The data presented differently (male vs. female of each substrain) and in other cases they were combined. It is better to present them consistently, i.e., present them as combined or male and female of each substrain.
  2. Only female mice of the two substrains are differents in active nose pokes but the authors indicated there was a difference in male and female of one substrain compared to the other. The authors claim is only true for "days to criteria".
  3. The same is true for the differences between two substrains. Only female mice B6J display more nose pokes than female B5NJ but the males are not different.
  4. There is no dose-response of the inhibitory effect of A33 on methamphetamine self-administration, all doses reduce methamphetamine self-administration to the same extent and there is no difference in the extent of inhibition between the low dose of A33 and other doses.
  5. Some information is not consistent between different section of the manuscript. For example, the highest dose of methamphetamine in the abstract is stated to be 1.6g/L but in the text and schematic diagram, it is stated 3.2 g/L. Similarly, the concentration of DMSO between the Results and Methods section varies, 10% vs. 0.1%.
  6. If the intake is the same, and the nose poke different, what does that mean?

Minor:

Introduction

  1. Is it addiction from the drug or the drug per se is a prevalent public health issue?
  2. Page 2, 4th paragraph, line 4, please change "then" to "than".
  3. Page 3, Line 6, Is it "exhibit for greater" or "exhibit far greater"? Please correct is needed.
  4. Page 7, last paragraph, line 1, is it 100 mg/L or 400 mg/L? Please correct is needed.
  5. Page 11, line 9, please change "a tolerance" to "tolerance".
  6. Page 11, something is missing in the last sentence of Paragraph one.
  7. Page 11, Section 3.3., Please connect lines 10 and 11. 

Author Response

Reviewer 1: We thank this reviewer for their kind words regarding our interesting findings and their efforts in reviewing our report.  Please find below our responses to their specific inquiries.

Major:

  1. The data presented differently (male vs. female of each substrain) and in other cases they were combined. It is better to present them consistently, i.e., present them as combined or male and female of each substrain.

Reply:  The specific data presented in each figure are intended to visually depict the statistical results for each dependent variable.  Thus, for results for which we detected a significant Substrain effect or interaction with the Substrain factor, the data are presented separately for each substrain.  For results for which there was no Substrain effect, the data were collapsed across the Substrain factor to highlight other significant interactions or main effects.  The same holds for the Sex factor.  If we were to always present the data as collapsed across an independent variable for which we detected a main effect or interaction, then that effect or interaction would be lost in the graphical depiction.  On the other hand, if we were to present the data consistently for both male and female subjects, even when there is no Sex effect or interaction, then number of panels in each figure would be overwhelming, obscuring any Substrain or Dose/Concentration effect that was detected.  To facilitate interpretation of the data, we worked hard to ensure that when sexes were combined, we used distinct colors for each symbol when the data were combined within sex (yellow vs. green squares) vs. when each sex was presented separately (blue triangles vs. red circles) and we indicated which independent variable was collapsed in both the text and in each figure panel to guide the reader.  Given the large amount of data presented in this report and the fact that the other reviewer had no issue with how the data were presented, we have opted to leave the figure formatting as is.      

  1. Only female mice of the two substrains are different in active nose pokes but the authors indicated there was a difference in male and female of one substrain compared to the other. The authors claim is only true for "days to criteria".

Reply: This reviewer is correct in that females exhibited signs of higher MA reinforcement than males only during the acquisition phase of testing (see Fig.1).  Thus, we have modified all text related to sex differences in MA reinforcement to highlight that sex differences were detected initially but did not hold up throughout the entire study (see end of Introduction, page 3).   

  1. The same is true for the differences between two substrains. Only female mice B6J display more nose pokes than female B5NJ but the males are not different.

Reply:  We are not completely clear to which data the reviewer is referring in this statement.  True, the data from late acquisition (Fig.1) suggest that the substrain difference in MA reinforcement is driven primarily by the females.  However, the statistical results of the analyses of the data in Fig.1 did not support a significant interaction between substrain and sex.  As such, only the significant main effects are highlighted in the figure.  As highlighted in Fig.2, B6J clearly exhibit more active hole poke and earn more reinforcers than B6NJ mice during A33 testing, but as discussed in the Discussion, these substrain effects disappear as the mice advanced through the study (see Figs. 3-6). In the hopes of addressing this reviewer’s concern, we have stipulated that substrain differences were also apparent during the earlier phases of the study (see end of Introduction, page 3). 

  1. There is no dose-response of the inhibitory effect of A33 on methamphetamine self-administration, all doses reduce methamphetamine self-administration to the same extent and there is no difference in the extent of inhibition between the low dose of A33 and other doses.

Reply: We agree that the dose-response function for A33 is rather flat and this finding is similar to what has been reported for A33’s effects on binge-drinking in our prior report.  However, results of within-subjects contrasts conducted on the data indicate linear effects of A33 pretreatment for active hole-responding, reinforcers earned and intake.  These additional statistical results are now included the manuscript in support of the weak, but statistically significant, dose-dependent effect of A33 on these variables.

  1. Some information is not consistent between different section of the manuscript. For example, the highest dose of methamphetamine in the abstract is stated to be 1.6g/L but in the text and schematic diagram, it is stated 3.2 g/L. Similarly, the concentration of DMSO between the Results and Methods section varies, 10% vs. 0.1%.

Reply:  We apologize for these typos.  We have corrected them throughout. 

  1. If the intake is the same, and the nose poke different, what does that mean?

Reply:  Nose-poking is an appetitive response, that we interpret as reflecting drug reinforcement or motivation to obtain drug, while the actual MA intake is a consummatory response, that we interpret as reflecting drug reward, but can be limited by a number of factors that are not readily discernable (e.g., satiety, aversiveness of drug experience, physical capacity to ingest large volumes of fluid). Thus, one explanation is that animals who want MA more will emit many nose-pokes, but may not have the physical capacity to consume all of the drug that was delivered.  A completely alternate explanation is that the MA consumed induced stereotypyed nose-poking behavior that has absolutely nothing to do with the reinforcing or rewarding properties of the drug.  The drug has merely engaged striatal circuits that generate nose-poking behavior.  However, arguing against this later interpretation/explanation, the dose-response functions for both active and inactive hole-poking peak at 200 mg/L.  If stereotypy was the sole explanation, then one would predict a more linear dose-response for nose-poking. 

Minor:

Introduction

  1. Is it addiction from the drug or the drug per se is a prevalent public health issue?

Reply:  Both are true as obviously addiction poses a significant health burden, but the drug itself poses significant criminal and legal burden and even recreational use of the drug in non-dependent individuals can feed into the criminal and legal burden of the drug itself.  We have revised the sentence in question to state “MA use” to cover all of these aspects

  1. Page 2, 4th paragraph, line 4, please change "then" to "than".
  2. Page 3, Line 6, Is it "exhibit for greater" or "exhibit far greater"? Please correct is needed.
  3. Page 7, last paragraph, line 1, is it 100 mg/L or 400 mg/L? Please correct is needed.
  4. Page 11, line 9, please change "a tolerance" to "tolerance".
  5. Page 11, something is missing in the last sentence of Paragraph one.
  6. Page 11, Section 3.3., Please connect lines 10 and 11. 

Reply:  We have corrected all of these typos and errors and we greatly appreciate the reviewer’s efforts in pointing them out.  Thank you.

Reviewer 2 Report

  • A significant portion of the introduction focuses on discussing neuroinflammation and emphasizes the role neuroinflammation may have in contributing to the neurotoxic and psychiatric effects of meth. However, the current report does not investigate neuroinflammation, neurotoxicity, or meth-induced psychosis. Even within the context of substrains differences, neuroinflammation is discussed. Neuroinflammation may play a role and the effect of PDE4B inhibition could be a result of anti-inflammatory effects but there are no experiments performed to address this.
  • Throughout the study proper controls have not been included. Separate groups allowed to nosepoke for water should be included to determine whether nosepoking behavior is driven by meth reinforcement or potentially by operant sensory stimulation. This is also important for understanding the sex and substrain differences presented in figure 2; is the reinforcing capacity of meth different between sexes and substrains or is there simply a difference in water consumption or operant sensory stimulation evoked by contingent presentation of the cue light/tone. Do mice self-administer when trained without contingently presented cue light/tone? These control experiments are particularly important given that there was no significant effect on nosepoking or # of MA reinforcers across the wide range of 25mg/L-3200mg/L in figures 3A and D.
  • Authors state in 2.1.3 that 400mg/L was used based on this dose being on the ascending limb of the dose-response from figure 3E; however, nosepoking behavior is largely independent of dose in the current study, again raising concerns about whether subjects are actually self-administering meth.
  • Behavioral data presented is inconsistent with the prior Ruan et al., JNS study wherein there was an approximate doubling of the number of nosepokes achieved with 80mg/L versus 160mg/L. In the JNS study 400mg/L produced on average >40 nosepokes whereas in the current report 400mg/L produce ~25 nosepokes. Similar discrepancies exist between MA intake although it is difficult to tell for concentrations below 200mg/L. These issues raise concerns regarding rigor and reproducibility.
  • In figure 1 legend and in the introduction the seeking test is referred to as ‘cue-induced reinstatement’ but a reinstatement paradigm is not used.
  • Based on figure 3 there is no substrain difference in self-administration behavior but in figure 5 there appears to be a difference or at least a strong trend at 0mg/kg A33; could this be a differential stress response to being injected? Also, if normalized to behavior at 0mg/kg A33 is the magnitude of effect for the A33 dose-response different?

Author Response

Reviewer 2:

  • A significant portion of the introduction focuses on discussing neuroinflammation and emphasizes the role neuroinflammation may have in contributing to the neurotoxic and psychiatric effects of meth. However, the current report does not investigate neuroinflammation, neurotoxicity, or meth-induced psychosis. Even within the context of substrains differences, neuroinflammation is discussed. Neuroinflammation may play a role and the effect of PDE4B inhibition could be a result of anti-inflammatory effects but there are no experiments performed to address this.

Reply:  We completely agree with this reviewer.  We were foreshadowing projects to come, rather than focusing on the study at hand.  We have modified the Introduction to focus on the potential clinical utility of PDE4B inhibitors, independent of their anti-inflammatory mechanism and we only mention inflammation once in the context of sex differences in MA-induced changes in PDE4 profiles as that was determined in the context of drug-induced inflammation. 

  • Throughout the study proper controls have not been included. Separate groups allowed to nosepoke for water should be included to determine whether nosepoking behavior is driven by meth reinforcement or potentially by operant sensory stimulation. This is also important for understanding the sex and substrain differences presented in figure 2; is the reinforcing capacity of meth different between sexes and substrains or is there simply a difference in water consumption or operant sensory stimulation evoked by contingent presentation of the cue light/tone. Do mice self-administer when trained without contingently presented cue light/tone? These control experiments are particularly important given that there was no significant effect on nosepoking or # of MA reinforcers across the wide range of 25mg/L-3200mg/L in figures 3A and D.

Reply:  One of the major goals of the original MA study was to examine for A33’s effects on cue-elicited responding during MA abstinence.  As such, each MA reinforcer was associated with a light/tone compound cue. In fact, we have never examined operant-conditioning for any reinforcer (alcohol, fentanyl, MA, oxycodone or sucrose) in the absence of cues as the data from the rat IV self-administration literature indicate that cues facilitate operant-learning, which is generally more difficult to perform in mice than in rats.  As such, we cannot comment on the relative influence of the cues vs. drug itself on active hole-poking. Indeed, the relatively flat MA dose-response function might suggest that the mice are responding primarily for the cues and such a suggestion is consistent with evidence that complex cues are reinforcing unto themselves in laboratory rodents. We have now highlighted this short-coming of our experimental design and mention “cue reactivity” as a potential confounding factor for our responding in our MA studies.

In all honesty, having conducted so many oral operant-conditioning studies in mice in recent years, we have noticed that, under procedures similar to those described for MA in this report, once mice have stabilized their responding during the acquisition phase of the operant-conditioning, they tend not to deviate in their responding, regardless of changes in drug dose or effort required for drug reinforcement.  This is apparent in the present study during the demand-response phase – the animal maintain their responding around 25 nose-pokes, regardless of the schedule of reinforcement.  This is approximately the same level of responding observed under an FR1 schedule during dose-response testing. At the present time, it is unknown if the number of responses would vary as a function of session duration or under a more traditional progressive ratio schedule where the response requirement changes rapidly within a session and these types of studies are currently planned in the laboratory. However, the fact that mice do not adjust their responding to compensate for increasing response requirements argues that (at least within the 1-h confines of the current procedures), mice are not necessarily responding for a particular number of reinforcers or cue presentations during the sessions.     

In considering this reviewer’s comment, we realized that we did fail to include a very important follow-up experiment to control for the effects of A33 on operant-responding.  While we had previously demonstrated that 1.0 mg/kg A33 did not reduce voluntary sucrose intake in the home cage, we do not know if A33 alters responding for a non-drug reinforcer under operant-conditioning procedures of relevance to the interpretation of the present results.  As we were limited in the amount of time allowed by IJMS to revise the report, we opted to conduct an additional study of A33’s effects on sucrose reinforcement in male and female B6J mice as we had published on sucrose reinforcement in male B6J mice previously.  While this new sucrose study cannot inform as to B6 substrain differences in sucrose reinforcement (which we have reported exist between 129 substrains), we detected higher sucrose intake by female vs. male B6J mice prior to A33 pretreatment, with no significant differences in active or inactive hole-poking. Such a results argues against at least a sex difference in mere responding for the light/tone cue.  Importantly for the interpretation of the A33 data, this new sucrose study showed that 0.03 mg/kg A33 (the lowest effective dose tested in the MA study) does not alter sucrose reinforcement, whereas the effect of 1.0 mg/kg A33 is non-selective. 

We hope this reviewer can appreciate the time-crunch imposed by the journal and is satisfied with the revisions to the Discussion pertaining to the issue of cue-reinforced responded, as well as the inclusion of the new data pertaining to sucrose reinforcement. 

  • Authors state in 2.1.3 that 400mg/L was used based on this dose being on the ascending limb of the dose-response from figure 3E; however, nosepoking behavior is largely independent of dose in the current study, again raising concerns about whether subjects are actually self-administering meth.

Reply:  Figure 3E summarizes the dose-intake function, upon which 400 mg/L lies towards the bottom of the ascending limb. As discussed above, we currently do not know why mice fail to adjust their active hole-pokes with changes in drug dose (Figure 3A), but their intake certainly does increase with increasing dose as they maintain their responding.  As we measure the amount of MA remaining in the receptacle, we verify whether or not mice are actually consuming the drug and how much they are consuming. Thus, we are confident that the mice are consuming the MA (particularly at lower doses as the receptacles are usually empty).

  • Behavioral data presented is inconsistent with the prior Ruan et al., JNS study wherein there was an approximate doubling of the number of nosepokes achieved with 80mg/L versus 160mg/L. In the JNS study 400mg/L produced on average >40 nosepokes whereas in the current report 400mg/L produce ~25 nosepokes. Similar discrepancies exist between MA intake although it is difficult to tell for concentrations below 200mg/L. These issues raise concerns regarding rigor and reproducibility.

Reply:  There are several procedural differences between the Ruan et al. JNS study and that present.  Firstly, Ruan et al. employed congenic B6 mice bred in-house at UCSB versus the present study employed isogenic B6J and B6NJ from JAX.  Thus, major differences exist between the two studies with respect to genetic background, mouse origin, and transportation/relocation stress at the outset of the studies.  Second, operant-conditioning occurred under different MA concentrations and for different durations.  Herein, the mice were first trained on 100 mg/L MA until responding stabilized, whereas in Ruan et al., the mice were trained on 80 mg/L MA for 2 weeks regardless of stabilization of responding to capture genotypic differences in MA reinforcement.  Thirdly, the present study was conducted early during the dark phase of the circadian cycle (starting at 10:00 h), while that of Ruan et al. was conducted later in the dark phase (starting at 13:00 h). This difference alone could account for a higher level of nose-poking behavior.  Lastly, the studies were conducted in two distinct buildings (Psychology in Ruan et al., versus Bio II in the present study), nearly 5 years apart and were conducted by completely different experimental staff.  Despite these major procedural differences, the mice in Ruan et al. exhibited ~25 responses under the 80 mg/L dose (Figure 3A in Ruan et al.) and the MA intake of the mice in the Ruan et al. study was between 1.5 and 3 mg/kg, which is in line with what we observed in the present study. Moreover, in both studies, the dose-active poking function exhibited an inverted U-shape. This shape is more apparent in Ruan et al., which could relate to any or all of the procedural differences listed above. Thus, we disagree with this reviewer regarding rigor and reproducibility as there are similarities in findings between the two studies, despite the studies NOT being replicates. 

  • In figure 1 legend and in the introduction the seeking test is referred to as ‘cue-induced reinstatement’ but a reinstatement paradigm is not used.

Reply:  That was a typo and has been corrected.  Thank you for catching that error.

  • Based on figure 3 there is no substrain difference in self-administration behavior but in figure 5 there appears to be a difference or at least a strong trend at 0mg/kg A33; could this be a differential stress response to being injected?

Reply:  As there is no statistical difference between the substrains during A33 testing, we did not discuss the marginally higher responding in B6J mice during the cue test. It should be noted that prior to any A33 injections, B6J also responded more than B6NJ mice (Fig. 1), which cannot be explained by differential “injection stress reactivity”.  The non-significantly higher responding exhibited by B6J mice on the cue test could reflect greater responsivity to injection, but it could also reflect a greater incubation of cue-elicited responding.  Unfortunately, the experimental design does not allow us to tease apart these mechanisms.  

  • Also, if normalized to behavior at 0mg/kg A33 is the magnitude of effect for the A33 dose-response different?

Reply:  This is an excellent question and the answer is no. When the data for active nose-pokes is normalized to VEH, the Sex X Substrain ANOVA did not indicate any main effects or interactions (all p’s>0.10). This supports our conclusion that A33 is equally effective in B6J and B6NJ mice.     

Round 2

Reviewer 1 Report

The authors adequately addressed the concerns of the previous version. I have no further comments/concerns.

Reviewer 2 Report

the sucrose experiment is an improvement to the manuscript; in this reviewer's opinion including a group to determine whether mice are engaging in operant sensory stimulation independent of meth reinforcement would be highly valuable in future studies